# Promoting high-voltage stability through local lattice distortion of halide solid electrolytes

Zhenyou Song[1,9], Tengrui Wang[1,9], Hua Yang[2,3,9], Wang Hay Kan [2,3] ✉,
Yuwei Chen[1], Qian Yu[1], Likuo Wang[1], Yini Zhang[1], Yiming Dai[1], Huaican Chen[2,3],
Wen Yin[2,3], Takashi Honda [4,5], Maxim Avdeev [6,7], Henghui Xu [8], Jiwei Ma [1],
Yunhui Huang [8] ✉ & Wei Luo [1] ✉

Stable solid electrolytes are essential to high-safety and high-energy-density lithium batteries, especially for applications with high-voltage cathodes. In such conditions, solid electrolytes may experience severe oxidation, decomposition, and deactivation during charging at high voltages, leading to inadequate cycling performance and even cell failure. Here, we address the high-voltage limitation of halide solid electrolytes by introducing local lattice distortion to confine the distribution of $Cl^-$, which effectively curbs kinetics of their oxidation. The confinement is realized by substituting In with multiple elements in $Li_3InCl_6$ to give a high-entropy $Li_{2.75}Y_{0.16}Er_{0.16}Yb_{0.16}In_{0.25}Zr_{0.25}Cl_6$. Meanwhile, the lattice distortion promotes longer Li-Cl bonds, facilitating favorable activation of $Li^+$. Our results show that this high-entropy halide electrolyte boosts the cycle stability of all-solid-state battery by 250% improvement over 500 cycles. In particular, the cell provides a higher discharge capacity of 185 mAh g$^{-1}$ by increasing the charge cut-off voltage to 4.6 V at a small current rate of 0.2 C, which is more challenging to electrolytes| cathode stability. These findings deepen our understanding of high-entropy materials, advancing their use in energy-related applications.

Advances in the development of lithium-ion batteries (LIBs) have revolutionized portable electronics and significantly changed our daily lives[1–3]. Although the applications of LIBs are extremely promising in electric vehicles, safety concerns have been raised due to the highly flammable liquid electrolyte used in LIBs, which could be quite dangerous upon thermal runaway. Solid electrolytes (SEs) present a promising solution to these safety concerns, as they are a group of nonflammable super ionic conductors with exceptionally high thermal stability and mechanical strength[4,5]. Therefore, SEs are widely believed to be capable of preventing fires or explosions. Among them, halide SEs have garnered significant interest recently due to superior Li$^+$ conductivity and decent electrochemical compatibility with cathode materials[6,7]. Specifically, a family of halide SEs ($Li_aMX_b$, M = Y, Er, Yb, In, Sc, Zr, etc. and X = F, Cl or Br) have emerged as an encouraging

[1]Institute of New Energy for Vehicles, School of Materials Science and Engineering, Tongji University, Shanghai 201804, China. [2]Spallation Neutron Source Science Center, Dongguan, Guangdong 523803, China. [3]Institute of High Energy Physics, Chinese Academy of Sciences, Beijing 100049, China. [4]Institute of Materials Structure Science, High Energy Accelerator Research Organization (KEK), Tsukuba, Ibaraki 305-0801, Japan. [5]J-PARC Center, High Energy Accelerator Research Organization (KEK), Tokai, Ibaraki 319-1106, Japan. [6]Australian Nuclear Science and Technology Organisation (ANSTO), Lucas Heights, NSW 2234, Australia. [7]School of Chemistry, University of Sydney, Sydney, NSW 2006, Australia. [8]State Key Laboratory of Material Processing and Die & Mould Technology, School of Materials Science and Engineering, Huazhong University of Science and Technology, Wuhan, Hubei 430074, China. [9]These authors contributed equally: Zhenyou Song, Tengrui Wang, Hua Yang. ✉e-mail: jianhx@ihep.ac.cn; huangyh@hust.edu.cn; weiluo@tongji.edu.cn

research hotspot in the field of all-solid-state batteries (ASSBs)[8–14], notably the LaCl$_3$-based halide SE[15], which is compatible with lithium metal. These advancements have paved the way for practical applications in the development of safer and more efficient ASSBs[16], highlighting the importance of continued research in this area.

Despite significant progress has been made in halide SEs, many issues are needed to address for achieving broader applications[17–19], including phase instability, structural degradation and poor cycling stability[20–22]. In particular, the incorporation of both higher voltage and larger specific capacity is technically important to achieve higher energy density[23]. However, the present electrochemical windows of most halide SEs are not efficient enough for practical applications[24]. As voltage exceeds 4.3 V, chloride ions undergo severe oxidation, resulting in the degradation of halide SEs and a significant rise in the interfacial resistance between cathodes and halide SEs[25–27]. Thus, halide SEs must possess improved oxidation resistance, which in turn hinges on the rational design of halides[28–30]. In recent decades, high-entropy materials, the configurational entropy ($S_{config}$) of which is maximized by introducing multiple elements into a single-phase lattice, have showed exceptional chemical/physical properties that are absent in the conventional materials[31]. The high entropy stability effect, characterized by $S_{config} > 1.5\,R$, is demonstrated to elicit thermodynamically stable states and foster homogenous elemental distribution. Additionally, the difference of atomic radii in high-entropy materials can induce local lattice distortion, thereby changing the energy barrier of element migration[32]. This impact effectively regulates the kinetics of different elements within the structure, showcasing the potential of high-entropy materials in mitigating corrosion and enhancing material stability[33–35]. Particularly, the latest discovered high-entropy SEs provide a new point of view to develop advanced SEs[36–40].

In this work, we design a new high-entropy halide SE, where local lattice distortion has been introduced to regulate kinetics of chloride ions and lithium ions diffusion, thereby greatly improving its oxidation stability and Li$^+$ conductivity. Multiple dopants, including Y, Er, Yb, and Zr, are introduced into the In site of Li$_3$InCl$_6$ (LIC), resulting in a single-phase Li$_{2.75}$Y$_{0.16}$Er$_{0.16}$Yb$_{0.16}$In$_{0.25}$Zr$_{0.25}$Cl$_6$ (HE-LIC). As confirmed by neutron diffraction, pair distribution function and bond valence energy landscape analysis, local lattice distortion induced by solute atoms has a selective impact on ions distribution and diffusion. The range of chloride ions vibration is restricted due to the local lattice distortion, leading to a more confined distribution and a less kinetically favorable oxidation. Moreover, the Li-Cl interatomic distances are extended, thereby promoting the activation and conduction of lithium ions within the high-entropy SE. Promoted by this unique design, HE-LIC simultaneously exhibits an enhanced Li$^+$ conductivity and a broadened electrochemical window. When coupling the HE-LIC with LiCoO$_2$ cathode, the as-assembled all-solid-state cell can deliver a prominent cycling performance of 500 cycles with an 88.9% capacity retention within 2.5–4.2 V at 0.5 C. More importantly, the cell gives a higher discharge capacity of 185 mAh g$^{-1}$ and a 91.6% capacity retention after 50 cycles at 0.2 C when increasing the charge cut-off voltage to 4.6 V. Our findings demonstrate that the local lattice distortion can significantly improve the electrochemical performance of halide SEs, providing a promising avenue for future research in high-voltage ASSBs.

## Results and discussion

### Synthesis and characterization of HE-LIC
The high-entropy Li$_{2.75}$Y$_{0.16}$Er$_{0.16}$Yb$_{0.16}$In$_{0.25}$Zr$_{0.25}$Cl$_6$ (HE-LIC) was prepared by a conventional solid-state reaction method, involving a high energy ball milling (BM) process of the precursors, followed by a heat treatment process in Argon atmosphere. The corresponding $S_{config}$ value is calculated as 1.589 $R$, described in the Supporting Information. The pristine Li$_3$InCl$_6$ (LIC) and a mid-entropy Li$_{2.75}$Y$_{0.5}$In$_{0.25}$Zr$_{0.25}$Cl$_6$

(ME-LIC) were also synthesized through the same method for a systematic comparison. The HE-LIC is composed of primary particles with hundreds of nanometers in size (Supplementary Fig. 1). Compared to LIC, the HE-LIC has a smaller size. Moreover, the nanometer-scale distribution of each metallic element in the HE-LIC is uniform, as evidenced by energy-dispersive X-ray (EDX) spectroscopy utilizing high-angle annular dark-field scanning transmission electron microscopy (HAADF-STEM) in Supplementary Fig. 2, indicating the mechanochemical method can effectively achieve a homogeneous mixing and adequate reaction of precursors.

To facilitate structure characterization and analysis, the powder X-ray diffraction (PXRD) measurement was first conducted on a glass holder covered with a Kapton film, owing to the inherent moisture sensitivity of the halide SEs. However, the Kapton film unavoidably contributed a relatively large background between 10 to 30 degrees in the XRD patterns (Supplementary Fig. 3), which resulted in a great challenge to analyze their structural properties. In addition, elements with similar electron densities are difficult to be distinguished using X-ray scattering. Moreover, X-ray is less sensitive for lithium. As such, neutron diffraction (ND) and neutron total scattering were adopted to further characterize the long-range and short-range crystallographic properties of these halide SEs. The Rietveld refinements of the ND patterns are given in Fig. 1a, b and Supplementary Fig. 4 and detailed lattice parameters were summarized in Supplementary Table 1, 2, 3. The crystal structure of pristine LIC is confirmed to be a single phase with monoclinic space group of C2/m, in good agreement with previous reports[9,41]. With adding Y and Zr into the structure, the ME-LIC appeared to be a mixture. The major phase (63.0 wt.%) is isostructural with Li$_3$InCl$_6$ (C2/m structure). Pnma-type Li$_3$YCl$_6$ (37.0 wt.%) is found to be the minor phase. After further introducing Er and Yb, the resulting HE-LIC exhibits the monoclinic Li$_3$InCl$_6$ structure (C2/m) together with some rock-salt LiCl, but without the Pnma-type Li$_3$YCl$_6$ phase. The weighted R-patterns (R$_{wp}$) for HE-LIC, ME-LIC and LIC are 1.738%, 1.727% and 2.884%, respectively, indicating fitting's excellent agreement with the experimental data. These results indicate that the structure stability driven by high entropy could enhance the solid solubility of various cations into the Li$_3$InCl$_6$ framework. In the Supplementary Fig. 5, 6 and Supplementary Discussion 1, further investigation regarding the second phase (LiCl, 6.5 wt.%) in HE-LIC will be discussed, which is believed to form during post heat treatment. Additionally, the process of optimizing element proportion in HE-LIC is illustrated in Supplementary Figs. 7, 8 and Supplementary Discussion 2.

As schematically shown in Supplementary Fig. 4b, the lattice of Li$_3$InCl$_6$ can be indexed to the rock-salt structure in which Cl$^-$ ions form the face-centered cubic (FCC) framework where the octahedral sites were occupied by cations (i.e., In and Li) and vacancy. Since the valence states of In and Li ions are 3$^+$ and 1$^+$, 1/3 of the metal sites are vacant according to the charge neutrality. This effect reduces the symmetry from Fm-3m into C2/m. InCl$_6$, LiCl$_6$ and VCl$_6$ (where V represents vacancy) octahedra share edges with each other to form a 3D structure. Similarly, the structural arrangement of the HE-LIC involves the stacking of MCl$_6$ (M = Y, Er, Yb, In, Zr) octahedra and LiCl$_6$ octahedra in the framework of Li$_3$InCl$_6$, which are visually presented in Fig. 1c. Fast Li$^+$ ion conduction, therefore, is facilitated via a 3D pathway involving LiCl$_6$ octahedral sites as well as the VCl$_6$ at 2a and 4g positions. In the HE-LIC, the introduction of cations with different radii effectively induces lattice distortion, which could lead to an overlapping distribution of site energies for the Li ions and promote a percolating network with low activation energy[36]. Additionally, compared to the LIC, the HE-LIC appears to be less crystalline, as evidenced by its board diffraction peaks. This could be beneficial to interface stability as some less crystalline sulfide SEs were shown to have less side reactions with cathode materials during cycling for battery applications[42]. Overall, these findings confirm that multiple metallic elements are successfully

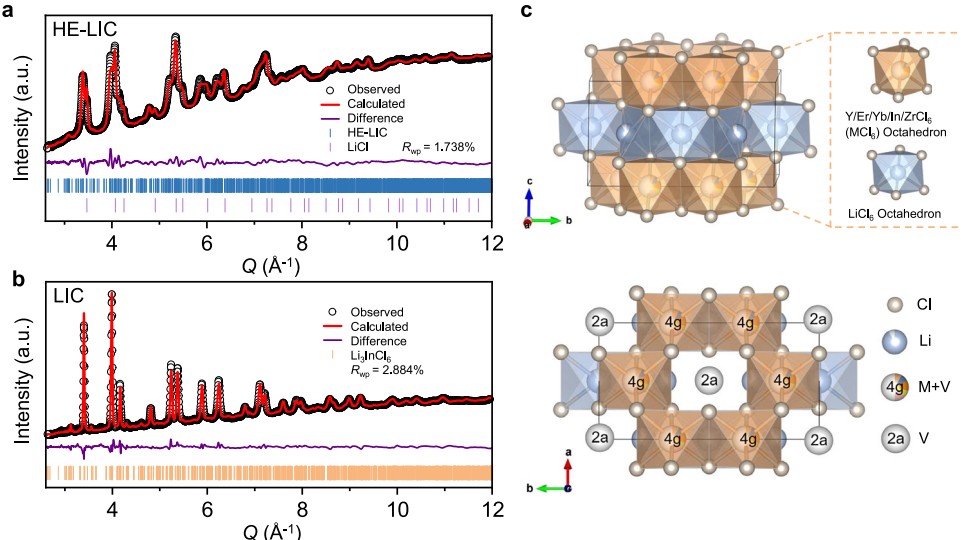

**Fig. 1 | Structure characterization and analysis.** Neutron diffraction patterns and the corresponding refinement of (**a**) HE-LIC and (**b**) LIC. **c** Graphical representations of a HE-LIC unit cell from the refinement, showing $MCl_6$ (M = Y, Er, Yb, In, Zr) octahedral and $LiCl_6$ octahedral framework.

incorporated into the unique glass-ceramic structure of the HE-LIC associated with the highest possible configurational entropy.

## Local structure exploration of HE-LIC

Subsequently, neutron total scattering was employed to analyze the local structures of the HE-LIC and LIC, as shown in Fig. 2. For a quick comparison, the pair distribution functions (PDF) for the HE-LIC and LIC are quite similar, indicating the crystal framework is identical, which is in good agreement with the ND results. Subtle differences were found especially for the individual peaks' intensity, which should be sensitive to the local atomic model. Typically, the nearest Li-Cl correlations are located at ~2.6 Å in Fig. 2a. It is noteworthy that the HE-LIC displays extended Li-Cl correlations, signifying an increased chance of Li-Cl breakage and higher possibility of Li+ activation. Then PDF refinements were carried out using supercells that allowed a single site in a unit cell to be occupied by only one type of atom based on the lowest coulomb energy configuration in Fig. 2b and Supplementary Fig. 9. The fitting results agree well with the experimental spectra in terms of peak positions and intensities, suggesting that the supercells could represent the actual scenario. In supercells, the typical Li-Cl interatomic distances were 2.6069 Å and 2.5818 Å for HE-LIC and LIC. Furthermore, a detailed examination of the supercells was conducted to investigate the unique clustering of ions at the *4g* site in the HE-LIC. The positions of ions at the *4g* site were summarized in Supplementary Table 4. The local structural analysis revealed that $Zr^{4+}$ is exclusively surrounded by trivalent ions as its nearest octahedrons, with a representative interatomic distance of 6.3983 Å (Supplementary Fig. 10). This specific local arrangement could be effective in maximizing the coulombic interaction between the positively and negatively charged ions while minimizing the repulsion energies between the same charged species. Therefore, the local clustering indicates the selective location of $Zr^{4+}$ in the structure. Then the Li-ion transport pathways were analyzed through the utilization of bond valence energy landscapes (BVEL), specifically an isosurface of the bond valence mismatch, which revealed that the three-dimensional Li+ channels were unimpeded (Supplementary Fig. 11). Similarly, the probability density isosurfaces of chloride ions are depicted in Fig. 2c, d and Supplementary Fig. 12, suggesting a dispersed vibrating range and a connected distribution of Cl− in LIC. This simulated outcome may forecast inferior oxidation performance of LIC, as a more linked distribution of chloride ions could lead to their more kinetically favorable oxidation. In other words, the implementation of the high-entropy configuration results in

the local lattice distortion, which effectively hinders the kinetics of chloride ions oxidation, ultimately leading to a significant improvement in the high-voltage stability.

## Electrochemical properties of HE-LIC

Cells, featuring a blocking electrode configuration, were utilized to determine the ionic conductivities, activation energies and electronic conductivities of both the as-prepared LIC and HE-LIC SEs. The electrochemical impedance spectroscopy (EIS) spectra presented in Fig. 3a enable the calculation of Li+ conductivities for LIC and HE-LIC, which are 0.849 and 1.171 mS cm−1, respectively. The enhancement in Li+ conductivity, through the substitution of multiple dopants, can be bifurcated into two aspects. On the one hand, the introduction of high-valence $Zr^{4+}$ ions could induce more lithium vacancies according to charge neutrality. On the other hand, the M-site disorder caused by the incorporation of multiple dopants increases the configuration entropy, leading to lattice distortions and Li+ sub-structure redistribution[43,44]. Further analysis of temperature-dependent Nyquist plots (Supplementary Fig. 13) leads to the Arrhenius plots (Fig. 3b), which reveal a calculated activation energy of 0.338 eV for the HE-LIC. This value is slightly smaller than that of the LIC (0.357 eV), indicating a reduced energy barrier of Li+ migration within the HE-LIC.

In addition to the ionic conductivity, an ideal SE requires a low electronic conductivity to prevent electrical leakage or short circuits. In Supplementary Fig. 14, the electronic conductivities ($\sigma_e$) of LIC and HE-LIC, determined by d.c. polarization, are $7.300 \times 10^{-9}$ and $2.467 \times 10^{-9}$ S cm−1, respectively. These values are approximately six orders of magnitude lower than their respective Li+ conductivities, thereby confirming that Li+ was the only charge carrier upon working in a cell. All electrochemical properties and associated errors are summarized in Fig. 3c. Additionally, cyclic voltammetry (CV) was conducted to test the electrochemical stability window (ESW) using Li-In | $Li_6PS_5Cl$ | halide SE | halide SE-VGCF cells, where halide SE was mixed with 10 wt% vapor-grown carbon fiber (VGCF) as the working electrode. To avoid reduction of the halide SEs, a thin layer of $Li_6PS_5Cl$ (LPSCl) was sandwiched between the Li-In anode and the halide SE (Supplementary Figs. 15, 16 and Supplementary Discussion 3). As shown in the CV curves (Fig. 3d), the oxidation potentials of LIC and HE-LIC are found to be approximately 4.25 V and 4.4 V, respectively. More importantly, the HE-LIC based cell exhibited a significant decline in current value, indicating impeded kinetics of Cl− oxidation. Notably, the CV curve of the LIC based cell exhibited an additional oxidation peak around 3.9 V,

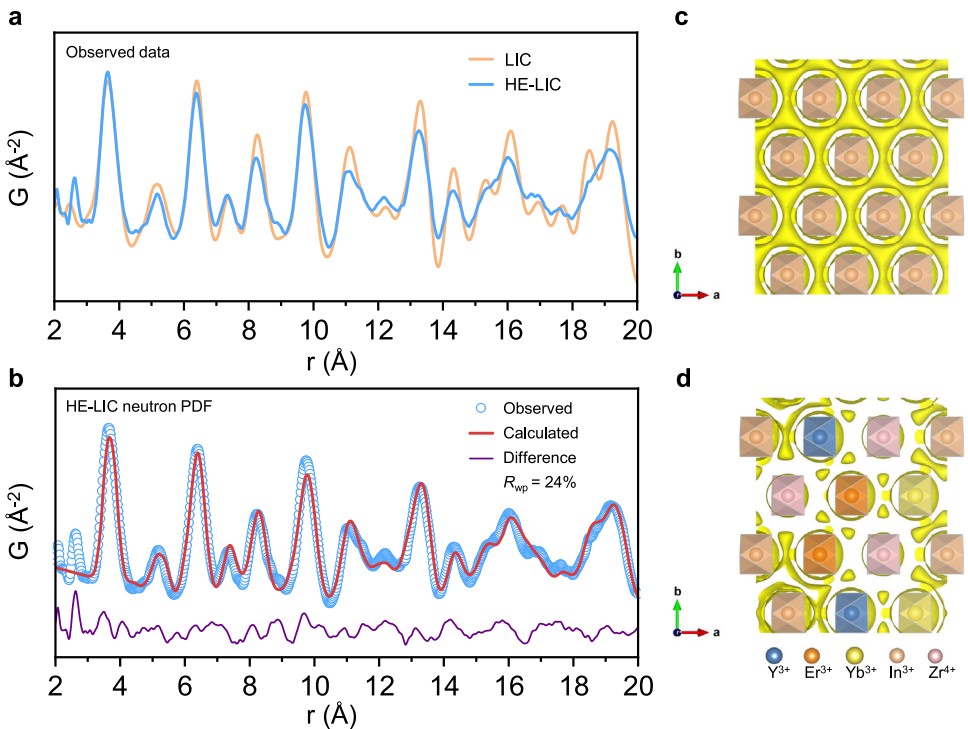

**Fig. 2 | Local structure in the HE-LIC. a** Neutron PDF data of LIC and HE-LIC between 2 Å and 20 Å. **b** PDF refinement of the HE-LIC for the atomic paired radial distribution with the lowest energy supercell as the structural model. **c, d** The probability density isosurface of chloride ions of the supercell based on BVEL: **c** LIC and **d** HE-LIC.

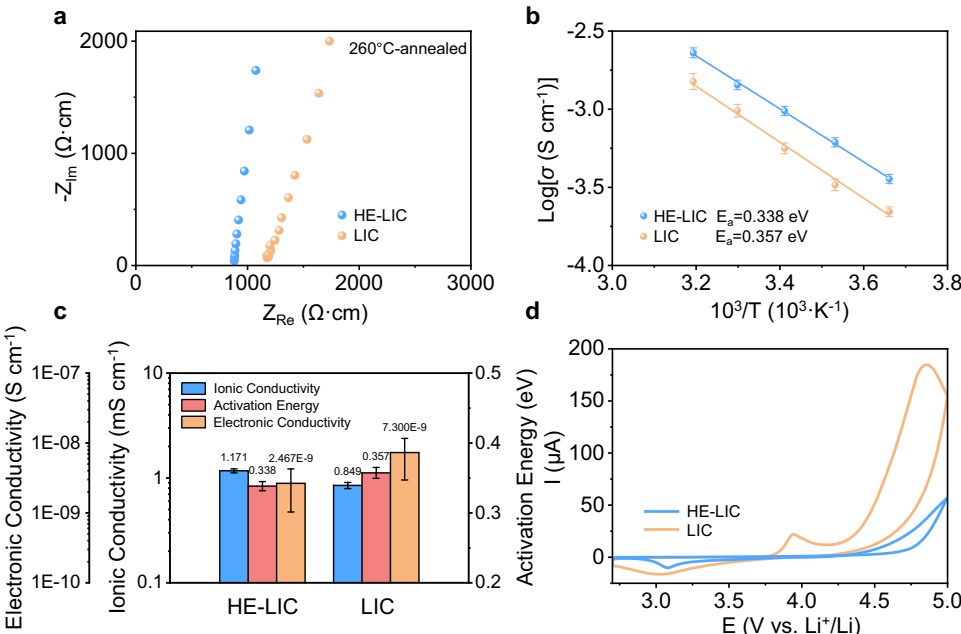

**Fig. 3 | The electrochemical properties and electrochemical windows of LIC and HE-LIC. a** Typical Nyquist plots at room temperature, normalized for the pellet thickness and area. **b** Arrhenius conductivity plots. **c** Summary of electrochemical properties. **d** CV curves of Li-In | LPSCl | halide SE | halide SE-VGCF cells within 2.7 and 5.0 V vs. Li⁺/Li at a scanning rate of 0.1 mV s⁻.

consistent with previous works[45]. As this oxidation peak is considered to be a self-limiting decomposition process, the resulting effect is a minor increase in impedance. In addition, we also measured the electrochemical properties of ME-LIC, which shows a Li⁺ conductivity of 0.930 mS cm⁻¹ and an activation energy of 0.341 eV (Supplementary Figs. 17, 18 and Supplementary Discussion 4). Overall, the above results indicate the introduction of local lattice distortion effectively

impacted the redistribution of Li⁺ and Cl⁻ sub-structure, leading to the improved ionic conductivity and oxidation stability of the HE-LIC.

## Electrochemical performance of ASSBs

In order to demonstrate the electrochemical performance of this new high-entropy halide SE, a layer of the HE-LIC was utilized as the separator to assemble the Li-In | LPSCl | HE-LIC | LCO cell (Fig. 4a),

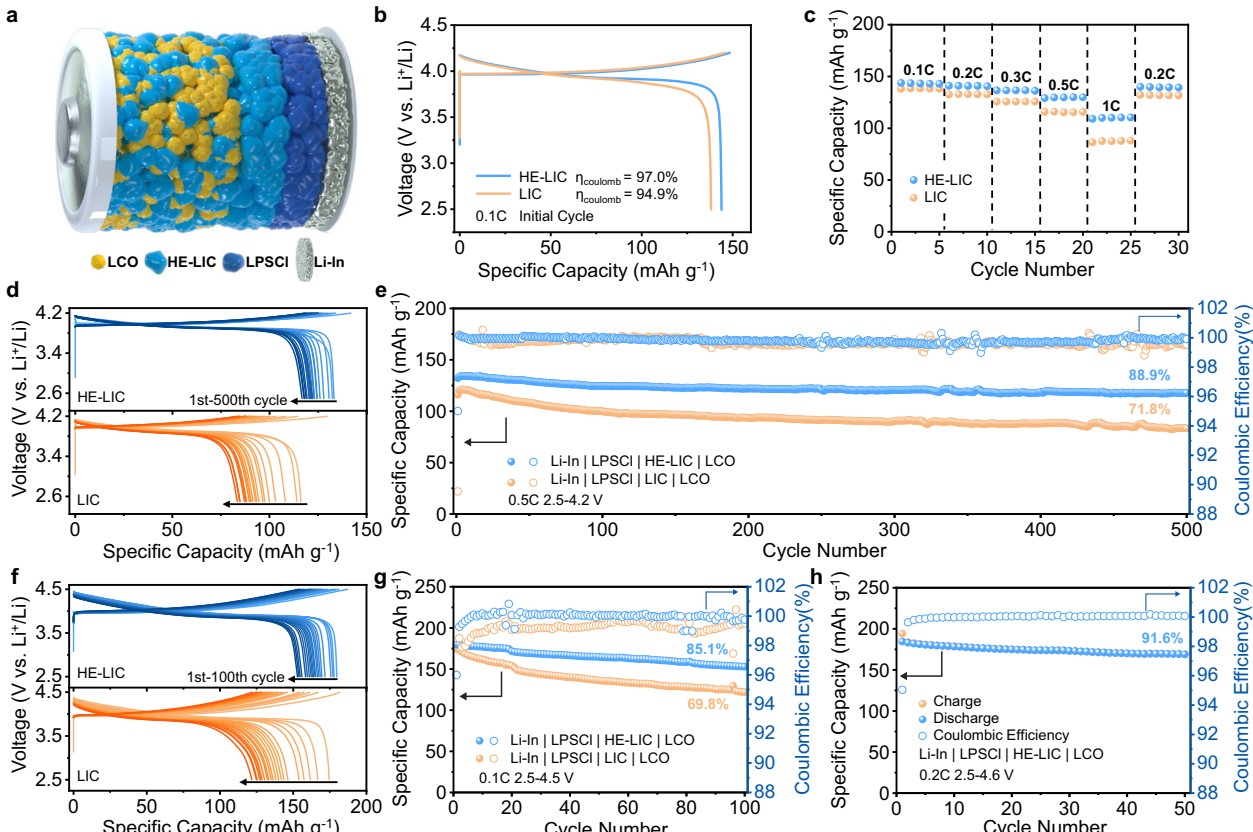

**Fig. 4 | Electrochemical performance of the all-solid-state batteries (ASSBs) by pairing halide SEs with LCO cathodes at room temperature. a** Schematic diagram of the cell configuration when the HE-LIC is used as SE. **b** The charge−discharge curves of cells tested at 0.1 C. **c** Rate capability of cells at 0.1, 0.2, 0.3, 0.5 and 1 C. **d**, **e** Charge-discharge curves and the corresponding long-term cycling performance of the cells at 0.5 C within a voltage window of 2.5–4.2 V. **f**, **g** Charge-discharge curves and the corresponding long-term cycling performance of the cells at 0.1 C within a voltage window of 2.5–4.5 V. **h** Cycling performance of the HE-LIC cell at 0.2 C within a voltage window of 2.5–4.6 V.

where the cathode is prepared by mixing $LiCoO_2$ (LCO) with HE-LIC at a weight ratio of 7:3 to facilitate sufficient Li⁺ conducting pathways. The 1st cycle galvanostatic charge−discharge voltage profile at 0.1 C (1 C = 140 mA g⁻¹) reveals that the HE-LIC enables a specific discharge capacity of 144 mAh g⁻¹ and an initial Coulombic Efficiency (ICE) of 97.0% (Fig. 4b), while the cell with the LIC delivers a lower capacity of 138 mAh g⁻¹ and a lower ICE of 94.9%. For the rate performance, the HE-LIC cell exhibits specific discharge capacities of 144, 141, 137, 129, and 109 mAh g⁻¹ at various rates of 0.1, 0.2, 0.3, 0.5 and 1 C, respectively (Fig. 4c). Additionally, the specific discharge capacity of the HE-LIC cell can recover to 140 mAh g⁻¹ when the current density is returned to 0.2 C. In contrast, the LIC cell exhibits worse performance at each rate (138, 133, 126, 116, 86 mAh g⁻¹ at 0.1, 0.2, 0.3, 0.5 and 1 C, respectively). Furthermore, the long cycling stability is demonstrated in Fig. 4d and e. After 500 cycles at 0.5 C, the HE-LIC cell maintains a capacity retention of 88.9%, which is significantly higher than the LIC cell (71.8%).

To further validate the enhanced oxidation resistance of HE-LIC, the charging voltage was elevated to 4.5 V and the long-term cycling was tested at 0.1 C. Compared to the test at higher current densities, more side reactions would occur at a low current rate, thereby challenging the stability of the halide SEs. As shown in Fig. 4f and g, the HE-LIC cell displays a capacity retention of 85.1% after 100 cycles, while the LIC counterpart only retains 69.8%. More surprisingly, the HE-LIC cell can even endure a higher charge cut-off voltage of 4.6 V, which delivers an initial capacity as high as 185 mAh g⁻¹ and maintains a capacity retention of 91.6% with an average CE of 99.9% over 50 cycles, suggesting its superior stability towards high voltage applications (Fig. 4h

and Supplementary Fig. 19). The improved rate performance and cycling life of the HE-LIC cell suggest that the HE-LIC not only offers a better Li⁺ conduction, but also guarantees the great stability at higher voltages, providing valuable insights for the design of high-performance all solid-state batteries.

## Post-cycling interfacial evolution

Post-cycling EIS measurements and X-ray photoelectron spectroscopy (XPS) characterizations were carried out to investigate the LCO|halide SE interfacial evolution, as depicted in Fig. 5. Notably, the semi-circular features observed at high-frequency (HF), mid-frequency (MF) and low-frequency (LF) correspond to the grain boundary resistance of SE ($R_{GB}$), charge-transfer resistance through cathode|SE interface ($R_{LCO|SE}$), and anode|SE interfacial resistance ($R_{Li-In|SE}$), respectively[46]. After 500 cycles, the charge-transfer resistances of the LCO | HE-LIC interface ($R_{LCO|HE-LIC}$) is only 47 Ω, almost three-fold lower than that of the LCO | LIC interface (150 Ω), as shown in Fig. 5a, b. This notable improvement can be attributed to the enhanced oxidation stability of HE-LIC. Detailed calculations and comparisons are compiled in Supplementary Table 5, 6. Additionally, due to the high-frequency limitation, the equivalent circuit of Li-In | LPSCl | HE-LIC | LCO is simplified to R(RQ)(RQ)Q as the semi-circle at HF cannot be detected. A similar phenomenon is observed in the impedance measurements before cycling in Supplementary Fig. 20, which shows the initial impedances of 75 Ω and 122 Ω for the HE-LIC cell and the LIC cell, respectively. The Cl 2p XPS spectra of the cathode in the HE-LIC and HIC cells are presented in Fig. 5c, d. For the HE-LIC cell, the 2p peaks display a similar shape to the pristine samples before cycling, with the exception of

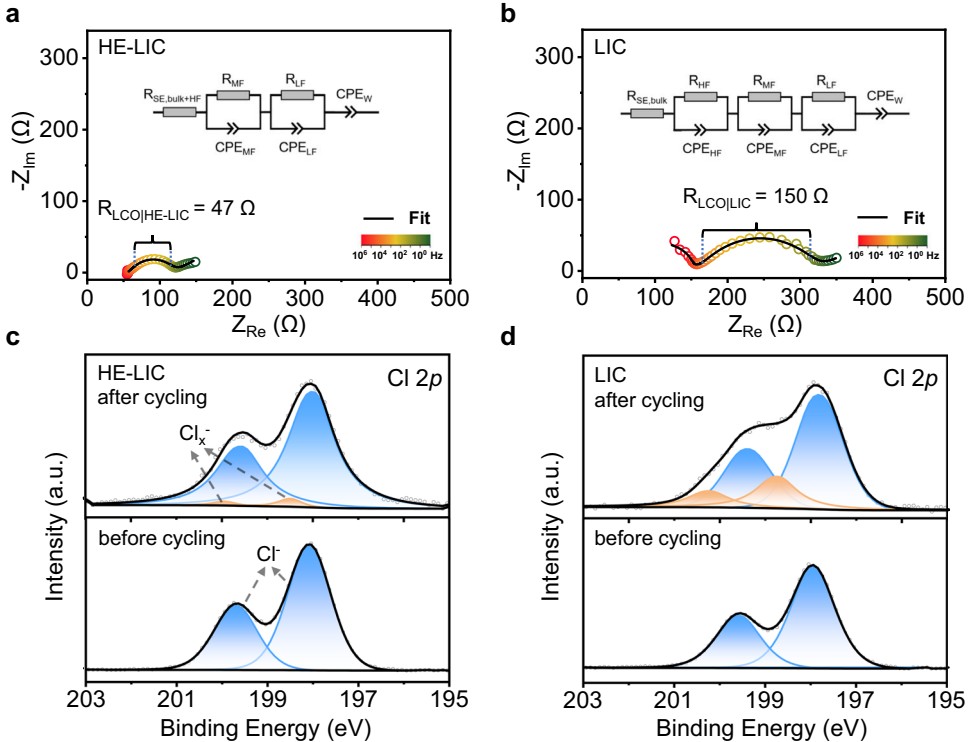

**Fig. 5 | Interfacial evolution of ASSBs. a, b** Impedance measurements from 1 MHz to 0.1 Hz and fitted results of ASSBs after cycling: **a** HE-LIC cell and **b** LIC cell. **c, d** Cl 2*p* XPS spectra of the cathode before and after cycling: **c** HE-LIC cell and **d** LIC cell.

some new peaks at higher binding energy after cycling, suggesting the presence of a few oxidation products of $Cl_x^-$, which correlates with the slightly increased interfacial impedance. Conversely, the LIC cell exhibits an obvious alteration in the shape of the Cl 2*p* peaks and generates broad peaks of $Cl_x^-$ after cycling, resulting in the significant rise in interfacial impedance and capacity attenuation. These results confirm that the HE-LIC cell maintained a more stable interface over cycling, making the HE-LIC a promising candidate for advanced energy storage applications.

In addition, we substituted the rare earth elements Y, Er, and Yb with trivalent elements Al, Ti, and Fe to synthesize a new HE electrolyte, $Li_{2.75}Al_{0.16}Ti_{0.16}Fe_{0.16}In_{0.25}Zr_{0.25}Cl_6$. The XRD analysis, as depicted in Supplementary Fig. 21 and Supplementary Discussion 5, reveals that this new compound also belongs to the monoclinic space group of C2/m. Despite exhibiting slightly inferior electrochemical performance (Supplementary Figs. 22, 23), the compositional flexibility of this material offers a promising solution to mitigate the industry's dependence on a single source of critical metals or rare earth elements. This approach presents an effective strategy for reducing the production cost of halide SEs by utilizing cost-effective elements, thereby facilitating the widespread implementation of halide SEs in practical applications.

In summary, we have successfully synthesized a high-entropy halide SE, $Li_{2.75}Y_{0.16}Er_{0.16}Yb_{0.16}In_{0.25}Zr_{0.25}Cl_6$, through the mechanochemical method. The unique glass-ceramic structure of the HE-LIC has been thoroughly characterized by ND, PDF and BVEL analysis, revealing that the high-entropy effect results in local lattice distortion and shorter Li-Cl bonds as well as a confined chloride ions distribution. These distinctive local structural features selectively impacted the kinetics of $Li^+$ and $Cl^-$ diffusion, leading to the improved electrochemical properties of HE-LIC, including $Li^+$ conductivity, oxidation resistance and cycling stability. Thereby the performance of ASSBs based on the HE-LIC is remarkably competitive with the reported halide-based ASSBs, as summarized in

Supplementary Table 7[11,28,30,47-49]. Furthermore, the HE-LIC can serve as a platform to further research into the cations disorder that governs the conduction of $Li^+$ ions and the role of the M-site in oxidation stability of $Li_aMX_b$[50-52]. Overall, our findings open up more compositional possibilities to discover novel SEs with optimized properties, and reveal further applications of high-entropy materials in the field of batteries.

## Methods

### Materials synthesis

The preparation of all compounds was conducted under an argon (Ar) atmosphere. LiCl (99%, Aladdin), YCl₃ (99.95%, Aladdin), ErCl₃ (99.9%, Aladdin), YbCl₃ (99.9%, Aladdin), InCl₃ (99.99%, Aladdin) and ZrCl₄ (99.9%, Aladdin) were used as received. For preparing high-entropy $Li_{2.75}Y_{0.16}Er_{0.16}Yb_{0.16}In_{0.25}Zr_{0.25}Cl_6$ (HE-LIC), about 2 g precursors were weighed according to the chemical formula and ground in a mortar evenly for 15 min. Then the precursors were subjected to ball milling in a ZrO₂ pot with ZrO₂ balls at 550 rpm for 30 h. The mass ratio of balls to precursors was 30:1, and the milling process involved alternating periods of 15 min of ball milling followed by 5 min of rest. Subsequently, the mixture was pelletized and annealed at 260 °C for 5 h with heating rate of 2 °C min⁻¹ under Ar atmosphere[9]. The pristine $Li_3InCl_6$ (LIC) and mid-entropy $Li_{2.75}Y_{0.5}In_{0.25}Zr_{0.25}Cl_6$ (ME-LIC) were prepared similarly. Argyrodite $Li_6PS_5Cl$ (LPSCl) electrolytes were synthesized as the reported articles[53].

### Material characterizations

X-Ray Diffraction (XRD) patterns were collected using a Bruker D8 ADVANCE diffractometer with Cu Kα radiation by measuring the diffraction angle range of 10–80°. TOF neutron diffraction and total scattering measurements were conducted at Multi Physics Instrument (MPI) in China Spallation Neutron Source (CSNS), and NOVA in Japan Proton Accelerator Research Complex (J-PARC). About 2–3 g of powders was put into Ti-Zr mull matrix alloy cans and the measurement

time was about 3 h for each sample. The diffraction datasets were subsequently analyzed by GSAS2[54]. The background in the neutron pattern was slightly higher for the high entropy solid electrolyte which was due to the presence of Cl, Yb, Li, Er, and In ions as they have the absorption cross section of 33.5, 34.8, 70.5, 159, and 193.8 barns respectively. On the other hand, Zr, Y, contribute relatively low background as they have low absorption cross section of 0.185 and 1.28, respectively. The occupancies of Er and In were only 16 mol.% and 25 mol.% in the 4 g sites (0, 0.3333, 0), therefore the general absorption of the sample was still acceptable for accurate structural analysis. The sample absorption parameter was calculated and manually input into the refinement process. To analyze the pair distribution function (PDF) datasets, different sizes of supercells were first created using SUPERCELL[55], by assigning the valence states of Li, Y, Er, Yb, In, Zr, and Cl as +1, +3, +3, +3, +3, +4, and −1. The columbic energies of the supercells were calculated; lowest energies supercells were selected as the input cif files to fit the PDF dataset using PDFGUI[56]. The lithium-ion and chloride-ion migration behaves were estimated by 3DBVSMAPPER[57]. Bond Valence Energy Landscape (BVEL) was calculated for lithium-ion and chloride-ion migration pathways[58]. The valence states of Li, Y, Er, Yb, In, Zr, and Cl were assigned as +1, +3, +3, +3, +3, +4, and −1. Vesta software was applied to visualize the crystal structure[59]. The morphology was investigated via scanning electron microscopy (SEM, sigma 300vp; Zeiss, Germany). The distribution of elements images was obtained using JEM-ARM300F STEM, equipped with integrated aberration (Cs) corrector. X-ray photoelectron spectroscopy (XPS) measurements were performed using American Thermo Fisher Scientific ESCALAB 250Xi with a monochromatic Al Kα source (1486.6 eV) and analyzed by Thermal Avantage. The spectra were normalized to adventitious carbon at 284.8 eV.

## Electrochemical measurements

The ionic conductivities of the prepared halide electrolytes were measured using cold pressed electrolyte pellets under 3 tons with a diameter of 10 mm. Stainless steel was then used as the ion-blocking electrodes at both sides of the pellets. Then Potentio Electrochemical Impedance Spectroscopy (PEIS) at an AC amplitude of 10 mV and a frequency from 1 MHz to 0.1 Hz was carried out on ion-blocking batteries to determine the temperature dependence of halide electrolytes at temperatures from 273 to 313 K with a Biologic workstation (VMP3). Direct current (DC) polarization measurements were conducted on the pellets with applied voltage of 1 V for 60 min to determine the electronic conductivities of the samples. Cyclic voltammetry (CV) at the voltage ranging from 2.7 to 5 V vs. Li$^+$/Li at a scan rate of 0.1 mV s$^{-1}$ was conducted on Li-In | LPSCl | halide SE | halide SE-VGCF cells at room temperature to obtain the oxidation potential of samples. 10 wt% vapor-grown carbon fiber (VGCF) was introduced to achieve sufficient interfacial contact.

## Assembly of ASSBs

The all-solid-state batteries (ASSBs) were fabricated using the prepared LIC and HE-LIC inside the inert argon atmosphere glovebox. To obtain the composite cathode, LiCoO$_2$ was mixed with halide electrolytes at a weight ratio of 7:3 and ground for 10 min. The ASSBs were assembled with a layer of halide electrolyte as the separator, paired with LiCoO$_2$ as the cathode and Li-In alloy as the anode. A thin layer of LPSCl was used to separate the halide electrolyte from the Li-In anode. First, the halide electrolyte (50 mg) was added into a polyether ether ketone (PEEK) cylinder die with a diameter of 10 mm and pressed under 2.5 tons. Then the LPSCI (30 mg) was added to the anode side. Next, about 8 mg composite cathode was added to the cathode side and pressed. Finally, In foil (100 μm) and Li foil (50 μm) were sequentially added to the anode side. The galvanostatic charge−discharge studies of cells were conducted on a Neware battery test system (CT-4008T-5V20mA-164, Shenzhen, China), within a potential window of 2.5 − 4.2 V vs Li$^+$/Li at

25 °C. High-voltage tests operated within a potential window of 2.5 − 4.5 V or 2.5 − 4.6 V vs Li$^+$/Li.

## Data availability

All data that support the findings of this study are provided within the paper and its Supplementary Information. All additional information is available from the corresponding authors upon reasonable request. Source data are provided with this paper. https://doi.org/10.6084/m9.figshare.24943011.

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

## Acknowledgements

This work is financially sponsored by National Natural Science Foundation of China (92163120, W.L.) and (11805034, U1930102, W.H.K.). The authors appreciate the beamtime in MPI and NOVA granted from China Spallation Neutron Source and Japan Proton Accelerator Research Complex, respectively. The neutron experiment at the Materials and Life Science Experimental Facility of the J-PARC was performed under a user program (Proposal No. 2022BF2101).

## Author contributions

Z.Y.S. and W.L. conceptualized the idea. Z.Y.S. synthesized the materials and carried out the characterizations and electrochemical tests with the assistance of T.R.W., Y.W.C., Q.Y., L.K.W., Y.N.Z., H.H.X. and J.W.M. W.H.K., H.Y., H.C.C., W.Y., T.H. and M.A. carried out the ND and neutron PDF measurements, analyzed the results, and performed the corresponding supercell calculation. Y.M.D. participated in the schematic diagram. The manuscript was written by Z.Y.S. and revised by W.L., W.H.K., and Y.H.H. All authors have discussed the results and approved the submission.

## Competing interests

The authors declare no competing interests.

## Additional information

**Peer review information** : *Nature Communications* thanks Seung Ho Choi, Emil Hanc and the other anonymous reviewer(s) for their contribution to the peer review of this work. A peer review file is available.

