## [Peer Review File · Nature Communications]

Promoting high-voltage stability through local lattice distortion of halide solid electrolytesREVIEWER COMMENTS

Reviewer #1 (Remarks to the Author):

The authors report about the successful synthesis of a chlorine-based high entropy solid electrolyte. They evaluated their local- and long-range crystal structure and electrochemical properties. Introducing high entropy in such class of materials seems to lead to better electrochemical stability and higher ionic conductivity. This study might pave the way for an extensive exploration of multielement substituted halide-based solid electrolytes. Therefore, the manuscript could potentially be accepted for publication in Nature Communications after a major revision.

In the introduction section and ex situ evaluation of the SSB cells (line 309 following), the authors state several times that chloride oxidation is a severe issue for chlorine-based solid electrolytes if high potentials are reached. Please discuss this in more detail, as it seems highly unusual. What species might be formed and are there references (also reference XPS spectra which have observed similar contributions in the Cl XPS spectra)?

Introduction sections Line 81 – give more appropriate recent references for HE SEs (<https://www.science.org/doi/10.1126/science.add7138>, <https://pubs.acs.org/doi/abs/10.1021/acsmaterialslett.2c00667>, <https://pubs.acs.org/doi/abs/10.1021/acsmaterialslett.1c00817>)

Usually, high entropy materials are designed via introducing equimolar compositions. Why did the authors choose especially this (non-equimolar) compositions? Are there issues with phase stability or others? This is an important point for other researchers to understand and follow up such work. If other compositions were unsuccessfully tried, this should be included in the manuscript and supporting information.

Please give standard deviations for the refinement tables in the supp info (if freely refined, or was everything fixed?)

Please give details for the refinement of the NPD data. Especially as some used isotopes/elements show strong neutron absorption cross section for neutrons, how was this issue treated during refining the diffraction data?

Reviewer #2 (Remarks to the Author):

The publication reports the enhancement of properties of Li electrolyte material based on Li_3InCl_6 through the incorporation of various dopants, resulting in the formation of a high-entropy material. However, the results obtained from described experiments do not seem to convincingly confirm the conclusions drawn by the authors of the work. Some issues within the study are outlined below:

1. The authors conclude that the enhancement in properties stems from localized lattice distortions, which arise due to the formation of a high-entropy material. However, due to the nearly identical ionic radii and the degree of oxidation of all the dopants, one may have doubts whether the change results only from the introduction of several different dopants into the structure. A study previously reported in doi.org/10.1021/acs.chemmater.1c01348, showcased a similar enhancement in ionic conductivity by doping Li_3InCl_3 with zirconium,

resulting in a conductivity increase to 1.2 mS/cm. The authors suggested that properties improvement was mainly due to an increased number of vacancies in the Li sublattice. This prior research seems to challenge the conclusions drawn in the present paper.

2. The authors of the work should show the results of the chemical composition analysis of the tested samples. It should be ruled out whether the increase in the stability of the material is not caused by the introduction of an admixture of oxygen to the material, for example as a result of a long grinding time. The authors did not provide a detailed description of the sample preparation process, but conventional practice in this type of study involves utilizing ZrO₂ vials and balls during grinding. Given the lengthy grinding duration employed in this study, the possibility of introducing extra ZrO₂ into the material becomes significant, which would be consistent with the formation of an additional LiCl phase. The effect of improving the stability and conductivity in lithium electrolytes by oxygen substitution has been previously described in the literature for other materials.

doi.org/10.1021/acs.chemmater.9b00505

3. The authors should present the results obtained for ME-LIC for comparison to show that the effect is not only due to the introduction of additional Er and Yb cations. In the current version, ME results include only SEM and XRD results, but conductivity and stability effects are not described.

4. The authors did not describe the influence of on the stability towards Li. No justification was provided for the use of an additional LPSCI layer for the construction of the cells.

5. No error bars are defined or presented in any relevant figures.

While the work, after making the necessary corrections, may be of interest to a battery field audience, it may not have the sufficient scientific impact in this field expected from this journal. Therefore, it is not recommended to publish the manuscript in Nature Communications.

Reviewer #3 (Remarks to the Author):

This work introduces a new concept of a high-entropy halide solid electrolyte containing seven components, denoted as HE-LIC (Li_{2.75}Y_{0.16}Er_{0.16}Yb_{0.16}In_{0.25}Zr_{0.25}Cl₆). The authors elucidate that local lattice distortion in the HE-LIC structure leads to increased ionic conductivities and oxidation stability. The structural characterization and analysis of HE-LIC with highly complex MCl₆ frameworks are carried out meticulously using neutron diffraction, TEM, EDX, STEM, and especially the calculation of configurational entropy.

However, I have several doubtful points regarding the high-entropy solid electrolyte developed in this paper. Firstly, the HE-LIC proposed by the authors comprises expensive rare earth materials (Y, Er, Yb). Therefore, the authors need to provide insight into how to design and synthesize high-entropy electrolytes containing cheaper and more abundant elements such as Mn, Fe, Ti, or similar alternatives. It's worth noting that the Li-ion conductivities of Li₃InCl₆ and HE-LIC are 0.85 and 1.15 mS⁻¹, respectively, with activation energies of 0.36 eV and 0.34 eV. What effect does a difference of 0.02 eV have on the cell performances? In my opinion, the conductivity and activation energy achieved may not significantly improve battery performances.

The clear advantage of HE-LIC, when compared to Li₃InCl₆, is its superior oxidation stability, as evidenced by the CV curve (Figure 3d) and XPS results (Figure 5c). However, the explanation for understanding this improved oxidation stability is not adequately provided. Why did the local lattice distortion hinder the electrochemical redox reaction of the HE-LIC at high voltage?

While the HE-LIC is undoubtedly interesting, it remains challenging to identify a broader

significance in designing advanced electrolyte compositions through this work. Further consideration for publication in a high-impact journal like Nature Communications should depend on addressing these comments.

Itemized list of responses to the reviewers' report

(Black italics: Reviewers' report; Blue type: Reply to the Reviewer)

Reviewer #1: The authors report about the successful synthesis of a chlorine-based high entropy solid electrolyte. They evaluated their local- and long-range crystal structure and electrochemical properties. Introducing high entropy in such class of materials seems to lead to better electrochemical stability and higher ionic conductivity. This study might pave the way for an extensive exploration of multielement substituted halide-based solid electrolytes. Therefore, the manuscript could potentially be accepted for publication in Nature Communications after a major revision.

Reply to the Reviewer: We highly appreciate the referee for his/her summaries and positive comments of our work. High-entropy halide electrolyte's combination of largely enhanced electrochemical stability and high ionic conductivity makes this work novel and attractive. Below are the point-by-point responses.

1. In the introduction section and ex situ evaluation of the SSB cells (line 309 following), the authors state several times that chloride oxidation is a severe issue for chlorine-based solid electrolytes if high potentials are reached. Please discuss this in more detail, as it seems highly unusual. What species might be formed and are there references (also reference XPS spectra which have observed similar contributions in the Cl XPS spectra)?

Reply to the Reviewer: The authors thank the reviewer for his/her important comments. As shown in **Figure R1**, we first calculated the electrochemical window of Li_3InCl_6 (LIC) based on Density Functional Theory (DFT) calculations using Material Project. The cyclic voltammetry (CV) was also used to ascertain the oxidation potential of chloride SSEs (**Figure R2**), which aligns with theoretical calculation result. Clearly, as we set the cut-off voltage close to the oxidation potential and the catalysis of cathode active material, LIC undergoes inevitable oxidation in practical batteries. This unavoidable oxidation would result in capacity decay during long-term cycling of solid-state cells with LIC (**Figure R3**). To clarify the oxidized compositions, X-ray photoelectron spectroscopy (XPS) analysis was conducted. As can be seen from **Figure R4a**, a higher binding energy of Cl 2p peak is observed after oxidation, which shows a similar result with previous studies (**Figure R5**)¹. The resulting decomposition products include LiClO_4 , InClO etc.^{2,3}

In sharp contrast, the as-designed $\text{Li}_{2.75}\text{Y}_{0.16}\text{Er}_{0.16}\text{Yb}_{0.16}\text{In}_{0.25}\text{Zr}_{0.25}\text{Cl}_6$ (HE-LIC) induces lattice distortion due to the atoms of different radii. The lattice distortion leads to sluggish diffusion, primarily attributed to the coordinated diffusion of multiple elements, particularly chlorine, during the oxidation process. As a result, HE-LIC exhibited improved oxidation stability (**Figure R2**) and long-term cycling performance (**Figure R3**). This result is substantiated by the XPS analysis conducted after cycling (**Figure R4b**), which revealed a diminished signal of oxidation products. These findings confirm the considerable potential of high-entropy materials in enhancing the stability of all-solid-state batteries.

Figure R1. Thermodynamic equilibrium voltage profile and phase equilibria of Li_3InCl_6 , which is derived from Density Functional Theory (DFT) calculations based on Material Project.

Figure R2. CV curves of LIC and HE-LIC within 2.7 and 5.0 V vs. Li^+/Li .

Figure R3. Long-term cycling performance of solid-state cells using LIC or HE-LIC SSEs between a voltage window of 2.5-4.2 V at 0.5 C.

Figure R4. Cl 2p XPS spectra of the cathode before and after cycling: **a** LIC cell and **b** HE-LIC cell.

Figure R5. Cl 2p spectra of the (halides + C) slice in the Li/Li₇P₃S₁₁/Li₂ZrCl_{6-4x}F_{4x} ($x = 0, 0.2$)/SS cells: Li₂ZrCl₆ and Li₂ZrCl_{5.2}F_{0.8}. Adopted from [Ref. R1].

[Ref. R1] Luo X, He X, Su H, Zhong Y, Wang X, Tu J. Effective regulation towards electrochemical stability of superionic solid electrolyte via facile dual-halogen strategy. *Chem. Eng. J.* **465**, (2023).

[Ref. R2] Kochetkov I, *et al.* Different interfacial reactivity of lithium metal chloride electrolytes with high voltage cathodes determines solid-state battery performance. *Energy Environ. Sci.* **15**, 3933-3944 (2022).

[Ref. R3] Zhou L, Zuo T, Li C, Zhang Q, Janek J, Nazar LF. Li_{3-x}Zr_x(Ho/Lu)_{1-x}Cl₆ Solid Electrolytes Enable Ultrahigh-Loading Solid-State Batteries with a Prelithiated Si Anode. *ACS Energy Lett.* **7**, 3102-3111 (2023).

2. Introduction sections Line 81 – give more appropriate recent references for HE SEs (<https://www.science.org/doi/10.1126/science.add7138>, <https://pubs.acs.org/doi/abs/10.1021/acsmaterialslett.2c00667>, <https://pubs.acs.org/doi/abs/10.1021/acsmaterialslett.1c00817>)

Reply to the Reviewer: By following the kind suggestion of the reviewer, we have added the related papers as Ref. 38, 39, and 40 in the revised manuscript.

Revised references:

Ref. 38. Li Y, *et al.* A lithium superionic conductor for millimeter-thick battery electrode. *Science* **381**, 50-53 (2023).

Ref. 39. Strauss F, *et al.* High-entropy polyanionic lithium superionic conductors. *ACS Mater. Lett.* **4**, 418-423 (2022).

Ref. 40. Lin J, *et al.* A High-entropy multicationic substituted lithium argyrodite superionic solid electrolyte. *ACS Mater. Lett.* **4**, 2187-2194 (2022).

3. Usually, high entropy materials are designed via introducing equimolar compositions. Why did the authors choose especially this (non-equimolar) compositions? Are there issues with phase stability or others? This is an important point for other researchers to understand and follow up such work. If other compositions were unsuccessfully tried, this should be included in the manuscript and supporting information.

Reply to the Reviewer: We thank the reviewer for his/her in-depth comments.

As the reviewer mentioned, we have conducted investigation on the equimolar composition at first, and we should include this optimization content in the revised manuscript. The equimolar $\text{Li}_{2.8}\text{Y}_{0.2}\text{Er}_{0.2}\text{Yb}_{0.2}\text{In}_{0.2}\text{Zr}_{0.2}\text{Cl}_6$ (HE-LIC-equimolar), exhibits similar XRD patterns as the non-equimolar $\text{Li}_{2.75}\text{Y}_{0.16}\text{Er}_{0.16}\text{Yb}_{0.16}\text{In}_{0.25}\text{Zr}_{0.25}\text{Cl}_6$ (HE-LIC), as depicted in **Figure R6**. The HE-LIC-equimolar exhibits a Li^+ conductivity of 0.930 mS cm^{-1} and an activation energy of 0.352 eV (**Figure R7**). Furthermore, we have discovered that by slightly adjusting the proportion of certain elements on this basis, we can further enhance the electrochemical performance. For instance, incorporating more quadrivalent Zr results in additional lithium vacancies⁴. With more In, it can exhibit enhanced ionic conductivity in heat treatment compared to Y, Er, and Yb^{5,6}. Meanwhile, we have to ensure that the configurational entropy is not significantly compromised. Therefore, we reach the non-equimolar formula $\text{Li}_{2.75}\text{Y}_{0.16}\text{Er}_{0.16}\text{Yb}_{0.16}\text{In}_{0.25}\text{Zr}_{0.25}\text{Cl}_6$. As expected, this composition demonstrates an ionic conductivity of 1.171 mS cm^{-1} and a lower activation energy (0.338 eV) compared to HE-LIC-equimolar. They both show enhanced oxidation stability (**Figure R7d**). Therefore, we selected HE-LIC as the primary focus of this study.

We have added the related information in the revised manuscript and supporting information accordingly.

Figure R6. X-ray diffraction patterns of HE-LIC-equimolar, HE-LIC and LIC.

Figure R7. Electrochemical properties comparison of HE-LIC-equipolar, HE-LIC and LIC. **a** Typical Nyquist plots at room temperature, normalized for the pellet thickness and area. **b** Arrhenius conductivity plots. **c** Summary of electrochemical properties. **d** CV curves within 3.0 and 5.0 V vs. Li⁺/Li.

[Ref. R4] Park K-H, Kaup K, Assoud A, Zhang Q, Wu X, Nazar LF. High-Voltage Superionic Halide Solid Electrolytes for All-Solid-State Li-Ion Batteries. *ACS Energy Lett.* **5**, 533-539 (2020).

[Ref. R5] Schlem R, *et al.* Mechanochemical Synthesis: A Tool to Tune Cation Site Disorder and Ionic Transport Properties of Li₃MCl₆ (M = Y, Er) Superionic Conductors. *Adv. Energy Mater.* **10**, 1903719 (2020).

[Ref. R6] Li X, *et al.* Air-stable Li₃InCl₆ electrolyte with high voltage compatibility for all-solid-state batteries. *Energy. Environ. Sci.* **12**, 2665-2671 (2019).

4. Please give standard deviations for the refinement tables in the supp info (if freely refined, or was everything fixed?)

Reply to the Reviewer: By following the important suggestion, we have added the standard deviations for the revised refinement tables.

Revised Supplementary Table 1. Structural refinement details from the neutron diffraction data for HE-LiC.

$\text{Li}_{2.75}\text{Y}_{0.16}\text{Er}_{0.16}\text{Yb}_{0.16}\text{In}_{0.25}\text{Zr}_{0.25}\text{Cl}_6$ Space group: $C2/m$

$a = 6.223(5) \text{ \AA}$, $b = 11.141(6) \text{ \AA}$, $c = 6.414(7) \text{ \AA}$, $\beta = 108.25(2)^\circ$

Atom	x	y	z	Occupancy	Multiplicity	$U_{\text{iso}} (\text{\AA}^2)$
Cl1	0.266(2)	0	-0.254(2)	1	4	0.020(1)
Cl2	0.253(1)	0.172(1)	0.252(2)	1	8	0.020(1)
Y/Er/Yb/				0.08/0.08/0.08/		
	0	0.33330	0		4	0.033(4)
In/Zr				0.125/0.125		
Li1	0.5	0	0.5	0.9167	2	0.006(6)
Li2	0	0.204(3)	0.5	0.9167	4	0.006(6)

Revised Supplementary Table 2. Structural refinement details from the neutron diffraction data for ME-LiC.

$\text{Li}_{2.75}\text{Y}_{0.5}\text{In}_{0.25}\text{Zr}_{0.25}\text{Cl}_6$ Space group: $C2/m$

$a = 6.432(8) \text{ \AA}$, $b = 10.971(7) \text{ \AA}$, $c = 6.394(9) \text{ \AA}$, $\beta = 109.17(2)^\circ$

Atom	x	y	z	Occupancy	Multiplicity	$U_{\text{iso}} (\text{\AA}^2)$
Cl1	0.253(2)	0	-0.254(2)	1	4	0.024(2)
Cl2	0.251(1)	0.159(2)	0.247(3)	1	8	0.024(2)
Y/In/Zr				0.25/0.125/0.125		
	0	0.33330	0		4	0.023(3)
Li1	0.5	0	0.5	0.9167	2	0.110(7)
Li2	0	0.164(2)	0.5	0.9167	4	0.110(7)

Revised Supplementary Table 3. Structural refinement details from the neutron diffraction data for LIC.

Li_3InCl_6 Space group: $C2/m$

$a = 6.401(6) \text{ \AA}$, $b = 11.035(2) \text{ \AA}$, $c = 6.397(6) \text{ \AA}$, $\beta = 109.54(1)^\circ$

Atom	x	y	z	Occupancy	Multiplicity	$U_{\text{iso}} (\text{\AA}^2)$
Cl1	0.250(1)	0	-0.230(1)	1	4	0.0115(5)
Cl2	0.236(1)	0.1604(4)	0.243(7)	1	8	0.0115(5)
In1	0	0.33330	0	0.53	4	0.056(6)
Li1	0.5	0	0.5	1	2	0.104(9)
Li2	0	0.155(1)	0.5	1	4	0.104(9)

5. Please give details for the refinement of the NPD data. Especially as some used isotopes/elements show strong neutron absorption cross section for neutrons, how was this issue treated during refining the diffraction data?

Reply to the Reviewer: We thank the reviewer for his/her questions.

For the powder neutron diffraction measurement, about 2 g powdery samples were put into the Ti-Zr null matrix alloy holders. Each sample was measured for ca. 3 hours at multiple physics instrument at China Spallation Neutron Source (CSNS). The background in the neutron pattern was slightly higher for the high entropy solid electrolyte which was due to the presence of Cl, Yb, Li, Er, and In ions as they have the absorption cross section of 33.5, 34.8, 70.5, 159, and 193.8 barns respectively. On the other hand, Zr, Y, contribute relatively low background as they have low absorption cross section of 0.185 and 1.28, respectively. The occupancies of Er and In were only 16 mol.% and 25 mol.% in the 4g sites (0, 0.3333, 0), therefore the general absorption of the sample was still acceptable for accurate structural analysis. The sample absorption parameter was calculated and manually input into the refinement process.

We have added the related information in the revised manuscript.

Reviewer #2: The publication reports the enhancement of properties of Li electrolyte material based on Li_3InCl_6 through the incorporation of various dopants, resulting in the formation of a high-entropy material. However, the results obtained from described experiments do not seem to convincingly confirm the conclusions drawn by the authors of the work. Some issues within the study are outlined below.

Reply to the Reviewer: We thank the reviewer for the overall evaluations on our study. In response to the reviewer's concerns and make our discovery clearer, we have incorporated more experiments and analysis as below.

1. The authors conclude that the enhancement in properties stems from localized lattice distortions, which arise due to the formation of a high-entropy material. However, due to the nearly identical ionic radii and the degree of oxidation of all the dopants, one may have doubts whether the change results only from the introduction of several different dopants into the structure. A study previously reported in doi.org/10.1021/acs.chemmater.1c01348, showcased a similar enhancement in ionic conductivity by doping Li_3InCl_3 with zirconium, resulting in a conductivity increase to 1.2 mS/cm. The authors suggested that properties improvement was mainly due to an increased number of vacancies in the Li sublattice. This prior research seems to challenge the conclusions drawn in the present paper.

Reply to the Reviewer: We thank the reviewer for raising this question.

We agree with the reviewer that the introduction of Zr doping can enhance the ionic conductivity by creating more lithium vacancies. To address this query, we also synthesized Zr-doped LIC ($\text{Li}_{2.6}\text{In}_{0.6}\text{Zr}_{0.4}\text{Cl}_6$). The XRD analysis reveals that the crystal structure of $\text{Li}_{2.6}\text{In}_{0.6}\text{Zr}_{0.4}\text{Cl}_6$ aligns with that of LIC (**Figure R8**). As shown in **Figure R9**, $\text{Li}_{2.6}\text{In}_{0.6}\text{Zr}_{0.4}\text{Cl}_6$ exhibits an improved ionic conductivity (1.078 mS cm^{-1}) and a reduced activation energy of 0.342 eV compared to LIC (0.849 mS cm^{-1} and 0.357 eV), respectively, which is consistent with the reported article mentioned by the reviewer. We also added the article as Ref. 14 in the revised manuscript.

In this work, we would like to simultaneously improve the ionic conductivity and the oxidation stability of chloride SEs by introducing the lattice distortion. The $\text{Li}_{2.75}\text{Y}_{0.16}\text{Er}_{0.16}\text{Yb}_{0.16}\text{In}_{0.25}\text{Zr}_{0.25}\text{Cl}_6$ (HE-LIC) exhibits an even higher ionic conductivity and lower activation energy than $\text{Li}_{2.6}\text{In}_{0.6}\text{Zr}_{0.4}\text{Cl}_6$. This suggests that the improvement of HE-LIC's performance is not solely attributed to the increase in lithium vacancies resulting from Zr doping. Rather, it is also influenced by a significant presence of cation disorder defect^{7,8}. It has been proved that creating more configurational disorder within cation sites is an important strategy for promoting fast Li^+ migration in chloride SEs⁹. In our study, the cation disorder leads to the aggregated distribution of Cl^- , and elongates the Li-Cl bond. This structure in turn inhibits the binding of the anion framework to Li^+ , and enables the favorable Li^+ conduction. Moreover, $\text{Li}_{2.6}\text{In}_{0.6}\text{Zr}_{0.4}\text{Cl}_6$ exhibits an even lower oxidation potential than LIC, as revealed by the CV tests (**Figure R9**). This result indicates that Zr doping cannot enhance the high-voltage stability of halide solid electrolytes. Therefore, we found that the high-entropy halide solid electrolytes represent an innovative approach to enhance the electrochemical properties of halide solid electrolytes.

Figure R8. X-ray diffraction patterns of $\text{Li}_{2.6}\text{In}_{0.6}\text{Zr}_{0.4}\text{Cl}_6$, HE-LIC and LIC.

Figure R9. Electrochemical properties of $\text{Li}_{2.6}\text{In}_{0.6}\text{Zr}_{0.4}\text{Cl}_6$, HE-LIC and LIC. **a** Typical Nyquist plots at room temperature, normalized for the pellet thickness and area. **b** Arrhenius conductivity plots. **c** Summary of electrochemical properties. **d** CV curves of $\text{Li}_{2.6}\text{In}_{0.6}\text{Zr}_{0.4}\text{Cl}_6$, compared with HE-LIC and LIC.

[Ref. R7] Yu S, *et al.* Design of a trigonal halide superionic conductor by regulating cation order-disorder. *Science* **382**, 573-579 (2023).

[Ref. R8] Schlem R, *et al.* Mechanochemical Synthesis: A Tool to Tune Cation Site Disorder and Ionic Transport Properties of Li_3MCl_6 ($\text{M} = \text{Y}, \text{Er}$) Superionic Conductors. *Adv. Energy Mater.* **10**, 1903719 (2020).

[Ref. R9] Li X, *et al.* The Universal Super Cation-Conductivity in Multiple-cation Mixed Chloride Solid-State Electrolytes. *Angew. Chem. Int. Ed.* **62**, e202306433 (2023).

2. The authors of the work should show the results of the chemical composition analysis of the tested samples. It should be ruled out whether the increase in the stability of the material is not caused by the introduction of an admixture of oxygen to the material, for example as a result of a long grinding time. The authors did not provide a detailed description of the sample preparation process, but conventional practice in this type of study involves utilizing ZrO_2 vials and balls during grinding. Given the lengthy grinding duration employed in this study, the possibility of introducing extra ZrO_2 into the material becomes significant, which would be consistent with the formation of an additional $LiCl$ phase. The effect of improving the stability and conductivity in lithium electrolytes by oxygen substitution has been previously described in the literature for other materials.

doi.org/10.1021/acs.chemmater.9b00505

Reply to the Reviewer: The authors appreciate the reviewer's meaningful comments.

We fully agree with the reviewer's concern that the possible introduction of oxygen doping will improve the performance of solid electrolytes, which has been reported in relevant articles.

In this study, we prepared the LIC and HE-LIC with the same ball milling time (30 h). We first investigate the LIC sample using X-ray photoelectron spectroscopy (XPS). As shown in **Figure R10**, LIC did not exhibit discernible Zr 3d signal, which means that the ball milling procedure would not introduce ZrO_2 into the solid electrolytes. To further eliminate concerns about ZrO_2 doping, we employed inductively coupled plasma optical emission spectrometer (ICP-OES) to study the HE-LIC (**Table R1**), which gives the chemical formula as $Li_{2.797}Y_{0.160}Er_{0.159}Yb_{0.162}In_{0.254}Zr_{0.250}Cl_{6.002}$, almost the same as the ingredient proportion, indicating that no ZrO_2 -containing impurities are introduced during the ball milling process.

Besides, per the kind suggestions of the reviewer, we have described the synthesis process in detail in the revised manuscript as: "The preparation of all compounds was conducted under an argon (Ar) atmosphere. $LiCl$ (99%, Aladdin), YCl_3 (99.95%, Aladdin), $ErCl_3$ (99.9%, Aladdin), $YbCl_3$ (99.9%, Aladdin), $InCl_3$ (99.99%, Aladdin) and $ZrCl_4$ (99.9%, Aladdin) were used as received. For preparing high-entropy $Li_{2.75}Y_{0.16}Er_{0.16}Yb_{0.16}In_{0.25}Zr_{0.25}Cl_6$ (HE-LIC), about 2 g precursors were weighed according to the chemical formula and ground in a mortar evenly for 15 min. Then the precursors were subjected to ball milling in a ZrO_2 pot with ZrO_2 balls at 550 rpm for 30 h. The mass ratio of balls to precursors was 30:1, and the milling process involved alternating periods of 15 minutes of ball milling followed by 5 minutes of rest. Subsequently, the mixture was pelletized and annealed at 260 °C for 5 h with heating rate of 2 °C min^{-1} under Ar atmosphere. The pristine Li_3InCl_6 (LIC) and mid-entropy $Li_{2.75}Y_{0.5}In_{0.25}Zr_{0.25}Cl_6$ (ME-LIC) were prepared similarly. Argyrodite Li_6PS_5Cl (LPSCl) electrolytes were synthesized as the reported articles."

Figure R10. Zr 3d XPS spectra of LIC and HE-LIC.

Table R1. Stoichiometry of HE-LIC according to the ICP-OES analysis. Atomic ratios of Li, Er, Yb, In and Zr have been normalized by Y content.

Element	Mass ratio (%)	Atomic ratio
Li	5.4600	17.7836
Y	4.0004	1.0000
Er	7.4817	0.9939
Yb	7.8984	1.0147
In	8.1895	1.5855
Zr	6.4158	1.5632

3. The authors should present the results obtained for ME-LIC for comparison to show that the effect is not only due to the introduction of additional Er and Yb cations. In the current version, ME results include only SEM and XRD results, but conductivity and stability effects are not described.

Reply to the Reviewer: We appreciate the reviewer for his/her insightful questions.

We have carried out related experiments and added more evidence to address the reviewer's concerns. The XRD and ND refining results of ME-LIC are shown in **Figure R11**. Obviously, ME-LIC is not a single phase, which consists of a major phase (63 wt.%) isostructural with Li_3InCl_6 ($C2/m$ structure) and a distinct second phase (37

wt.%) characterized as *Pnma*-type Li_3YCl_6 . Furthermore, the neutron PDF analysis of the ME-LIC sample also confirmed this result (**Figure R12**). As illustrated in **Figure R13**, ME-LIC shows a Li^+ conductivity of 0.930 mS cm^{-1} , an activation energy of 0.341 eV and an electronic conductivity of $3.974 \times 10^{-9} \text{ S cm}^{-1}$. Furthermore, the CV test exhibits a comparable oxidation potential of 4.28 V as LIC (**Figure R14**). However, it should be noted that the electrochemical properties of ME-LIC may not accurately reflect the performance of the target product, $\text{Li}_{2.75}\text{Y}_{0.5}\text{In}_{0.25}\text{Zr}_{0.25}\text{Cl}_6$, due to the significant presence of this second phase.

We have added the relevant characterization results and discussion of ME-LIC in the revised supplementary information accordingly.

Figure R11. a X-ray diffraction patterns of LIC, ME-LIC and HE-LIC. b Neutron diffraction patterns and the corresponding refinements of ME-LIC.

Figure R12. PDF refinement of the ME-LIC for the atomic paired radial distribution with the lowest energy supercell as the structural model. A two-phase refinement approach was adopted, incorporating $Pnma$ -type Li_3YCl_6 .

Figure R13. Electrochemical properties of ME-LIC. **a** Typical Nyquist plots at room temperature, normalized for the pellet thickness and area. **b** Arrhenius conductivity plots. **c** DC polarization curves with an applied voltage of 1 V. **d** Summary of electrochemical properties.

Figure R14. CV curves of ME-LIC, compared with LIC and HE-LIC.

4. The authors did not describe the influence of on the stability towards Li. No justification was provided for the use of an additional LPSCl layer for the construction of the cells.

Reply to the Reviewer: We thanks for the reviewer's important questions.

To investigate the stability of halide solid electrolytes towards Li, we first performed CV tests on both LIC and HE-LIC in the voltage range of 0-3.0 V vs. Li^+/Li . **Figure R15** illustrates that both LIC and HE-LIC exhibit a weak cathodic peak at approximately 2.2 V. This peak corresponds to the initiation of a weak reduction process, specifically the conversion from In^{3+} to In^{2+} . Notably, when the voltage drops below 1.6 V, LIC undergoes a significant reduction reaction, whereas HE-LIC experiences reduction below 1.1 V. Furthermore, the reduction reaction current of HE-LIC is considerably lower than that of LIC, indicating that HE-LIC demonstrates improved reduction stability.

To further assess the reduction stability of LIC and HE-LIC, we assembled symmetric cells with Li-In anodes (**Figure R16**). The polarization voltage of the Li-In|LIC|Li-In symmetric cell exceeds 5 V, and the impedance exceeds 10000 Ω after just one cycle. HE-LIC exhibits improved cycling stability, as evidenced by a stable cycle lasting for 400 h. However, the polarization voltage remained above 0.8 V, and the impedance reached 4000 Ω after cycling, suggesting that HE-LIC is also unstable to Li-In. So, we have chosen $\text{Li}_6\text{PS}_5\text{Cl}$ (LPSCl) to prevent the reduction of halide electrolytes. Meanwhile, LPSCl enable a relatively low interface impedance with halide SEs¹⁰. It is believed that the combination of stable SEI-forming LPSCl together with halide SEs as cathode electrolyte may be a suitable solution in practice. In LPSCl based Li-In symmetric cells, it achieved a consistent and stable cycle lasting for 400 h. Notably, LPSCl demonstrated a polarization voltage below 0.01 V and an impedance of only 10 Ω . Moreover, per the kind suggestions of the reviewer, the relevant content has been added to the revised Supplementary Information.

Figure R15. CV curves of Li-In | LPSCl | halide SE | halide SE-VGCF cells within 0 and 3.0 V vs. Li⁺/Li at a scanning rate of 0.1 mV s⁻¹.

Figure R16. Cycling performance of Li-In symmetric cells based on LIC, HE-LIC and LPSCl, and corresponding post-cycle impedance spectra.

[Ref. R10] Riegger LM, Schlem R, Sann J, Zeier WG, Janek J. Lithium-Metal Anode Instability of the Superionic Halide Solid Electrolytes and the Implications for Solid-State Batteries. *Angew. Chem. Int. Ed.* **60**, 6718-6723 (2021).

5. No error bars are defined or presented in any relevant figures.

Reply to the Reviewer: By following the kind suggestion of the reviewer, we have added the error bars for **Revised Fig. 3**.

Revised Fig. 3. Electrochemical properties of LIC and HE-LIC. **a** Typical Nyquist plots at room temperature, normalized for the pellet thickness and area. **b** Arrhenius conductivity plots. **c** Summary of electrochemical properties. **d** CV curves of Li-In | LPSCI | halide SE | halide SE-VGCF cells within 2.7 and 5.0 V vs. Li⁺/Li at a scanning rate of 0.1 mV s⁻¹.

6. While the work, after making the necessary corrections, may be of interest to a battery field audience, it may not have the sufficient scientific impact in this field expected from this journal. Therefore, it is not recommended to publish the manuscript in *Nature Communications*.

Reply to the Reviewer: We thanks for the reviewer's comments.

The innovation of this article is that it is the first reported high-entropy strategy to improve oxidation stability of halide solid electrolyte, which is distinguish from previous studies that mainly focused on F doping or interfacial coating¹¹⁻¹³. This work is complemented by in-depth characterization via neutron diffraction and the pair distribution functions, showing the structure-property relationship with the increase in configurational entropy. Meanwhile, the impact of local lattice distortions on the distribution and diffusion of Li⁺ and Cl⁻ is thoroughly examined. Moreover, this work has the combined advantage of both improved ionic conductivity as well as electrochemical stability. Therefore, this study is enlightening the broad prospect of the application of high-entropy materials in the field of solid electrolytes.

[Ref. R11] Zhang S, *et al.* Advanced High-Voltage All-Solid-State Li-Ion Batteries Enabled by a Dual-Halogen Solid Electrolyte. *Adv. Energy Mater.* **11**, 2100836 (2021).

[Ref. R12] Yu R, *et al.* Manipulating Charge-Transfer Kinetics of Lithium-Rich Layered Oxide Cathodes in Halide All-Solid-State Batteries. *Adv. Mater.* **35**, 2207234 (2023).

[Ref. R13] Sun S, *et al.* Eliminating interfacial O-involving degradation in Li-rich Mn-based cathodes for all-solid-state lithium batteries. *Sci. Adv.* **8**, eadd5189 (2022).

Reviewer #3: This work introduces a new concept of a high-entropy halide solid electrolyte containing seven components, denoted as HE-LIC ($\text{Li}_{2.75}\text{Y}_{0.16}\text{Er}_{0.16}\text{Yb}_{0.16}\text{In}_{0.25}\text{Zr}_{0.25}\text{Cl}_6$). The authors elucidate that local lattice distortion in the HE-LIC structure leads to increased ionic conductivities and oxidation stability. The structural characterization and analysis of HE-LIC with highly complex MCl₆ frameworks are carried out meticulously using neutron diffraction, TEM, EDX, STEM, and especially the calculation of configurational entropy.

However, I have several doubtful points regarding the high-entropy solid electrolyte developed in this paper.

Reply to the Reviewer: We appreciate the reviewer for his/her overall summaries and comments of our work. In the subsequent sections, we provide responses to reviewer's questions.

1. Firstly, the HE-LIC proposed by the authors comprises expensive rare earth materials (Y, Er, Yb). Therefore, the authors need to provide insight into how to design and synthesize high-entropy electrolytes containing cheaper and more abundant elements such as Mn, Fe, Ti, or similar alternatives.

Reply to the Reviewer: We thank the reviewer for raising this question.

We totally agree with the reviewer that synthesizing solid electrolytes with cheaper and more abundant elements is meaningful. Thus, we substituted the rare earth elements Y, Er, and Yb with trivalent elements Al, Ti, and Fe to synthesize a new high-entropy (HE) electrolyte, $\text{Li}_{2.75}\text{Al}_{0.16}\text{Ti}_{0.16}\text{Fe}_{0.16}\text{In}_{0.25}\text{Zr}_{0.25}\text{Cl}_6$. The XRD analysis, as depicted in **Figure R17**, reveals that this new compound also belongs to the monoclinic space group of *C2/m*. Then, we further measure the other properties of $\text{Li}_{2.75}\text{Al}_{0.16}\text{Ti}_{0.16}\text{Fe}_{0.16}\text{In}_{0.25}\text{Zr}_{0.25}\text{Cl}_6$, which exhibits a Li^+ conductivity of 0.454 mS cm^{-1} , an activation energy of 0.395 eV and an electronic conductivity of $3.633 \times 10^{-9} \text{ S cm}^{-1}$ (**Figure R18**). Obviously, the new HE electrolyte based on more abundant elements shows a poor Li^+ conduction ability compared to the original Li_3InCl_6 (0.849 mS cm^{-1}) and the HE electrolyte based on rare earth elements (1.171 mS cm^{-1}).

We then also assembled Li-In | LPSCI | $\text{Li}_{2.75}\text{Al}_{0.16}\text{Ti}_{0.16}\text{Fe}_{0.16}\text{In}_{0.25}\text{Zr}_{0.25}\text{Cl}_6$ | LCO full cells. **Figure R19** presents the initial cycle charge-discharge curves, where the $\text{Li}_{2.75}\text{Al}_{0.16}\text{Ti}_{0.16}\text{Fe}_{0.16}\text{In}_{0.25}\text{Zr}_{0.25}\text{Cl}_6$ cell delivers an initial discharge capacity of 109.4 mAh g^{-1} and an initial Coulombic efficiency (ICE) of 91.5%. In comparison, the HE-LIC enables the cell of a higher discharge capacity and ICE value (132.3 mAh g^{-1} , 93.3%). Moreover, the original LIC SSE can also offer the cell a higher discharge

capacity of 116.2 mAh g⁻¹. Clearly, the lower capacity of the Li_{2.75}Al_{0.16}Ti_{0.16}Fe_{0.16}In_{0.25}Zr_{0.25}Cl₆ cell is limited by its lower Li⁺ conductivity.

Although the HE electrolyte based on more abundant and cost-effective elements is attractive, its electrochemical performance is worse than rare elements based HE-LIC. We hope the reviewer could kindly agree with us that the high-entropy concept is useful for the design of new solid-state electrolytes (SSEs), and the development of high-entropy SSEs and halide SSEs is still in their early stage. Further studies are undergoing in our lab to investigate the substitution of rare earth elements in halide SSEs.

Figure R17. X-ray diffraction pattern of Li_{2.75}Al_{0.16}Ti_{0.16}Fe_{0.16}In_{0.25}Zr_{0.25}Cl₆.

Figure R18. Electrochemical properties of Li_{2.75}Al_{0.16}Ti_{0.16}Fe_{0.16}In_{0.25}Zr_{0.25}Cl₆, compared with HE-LIC and LIC. **a** Typical Nyquist plots at room temperature, normalized for the pellet thickness and area. **b** Arrhenius conductivity plots. **c** Summary of ionic conductivity and activation energy. **d** DC polarization curve of Li_{2.75}Al_{0.16}Ti_{0.16}Fe_{0.16}In_{0.25}Zr_{0.25}Cl₆ with an applied voltage of 1 V.

Figure R19. The initial cycle charge–discharge curves of cells with three different halide SSEs tested at 0.5C.

2. It's worth noting that the Li-ion conductivities of Li_3InCl_6 and HE-LIC are 0.85 and 1.15 mS^{-1} , respectively, with activation energies of 0.36 eV and 0.34 eV . What effect does a difference of 0.02 eV have on the cell performances? In my opinion, the conductivity and activation energy achieved may not significantly improve battery performances.

Reply to the Reviewer: The authors appreciate the reviewer's meaningful comment.

We fully agree with the reviewer's opinion that solely enhancing Li^+ conductivity and reducing activation energy may not suffice to significantly improve the performances of the battery. The increased ionic conductivity can reduce the impedance of the battery and enable the cathode to deliver a higher capacity. Moreover, the high-voltage stability of the HE-LIC is playing a more important role in influencing the cycling stabilities of batteries. Specifically, LIC is prone to oxidation side reactions when charged to high voltage. Previous studies have addressed these challenges by incorporating F doping, which unfortunately compromises Li^+ conductivities^{14,15}. In this work, we adopt a high-entropy strategy to enhance high-voltage stability without compromising ionic conductivities. Specifically, by implementing this strategy, the Li-In | LPSCI | HE-LIC | LCO cells achieve an enhancement in the initial Coulombic Efficiency from 94.9% to 97% (**Figure R20a**). Consequently, when subjected to long-term cycling conditions, the capacity retention is significantly improved from 71.8% to 88.9% (**Figure R20b**).

Figure R20. a The initial cycle charge–discharge curves of cells tested at 0.1C. **b** Long-term cycling performance of the cells at 0.5 C.

[Ref. R14] Zhang S, *et al.* Advanced High-Voltage All-Solid-State Li-Ion Batteries Enabled by a Dual-Halogen Solid Electrolyte. *Adv. Energy Mater.* **11**, 2100836 (2021).
[Ref. R15] Sun S, *et al.* Eliminating interfacial O-involving degradation in Li-rich Mn-based cathodes for all-solid-state lithium batteries. *Sci. Adv.* **8**, eadd5189 (2022).

3. The clear advantage of HE-LIC, when compared to Li_3InCl_6 , is its superior oxidation stability, as evidenced by the CV curve (Figure 3d) and XPS results (Figure 5c). However, the explanation for understanding this improved oxidation stability is not adequately provided. Why did the local lattice distortion hinder the electrochemical redox reaction of the HE-LIC at high voltage?

Reply to the Reviewer: We appreciate the reviewer for his/her insightful questions.

The oxidation mechanism of LIC involves the electron-acquiring oxidation of Cl^- , the diffusion of unreacted Cl^- , and the coordinated diffusion of other elements leading to the formation of a new phase. During Cl^- diffusion, the diffusing ions undergo atomic migration by vacating their original sites and occupying vacancies, which is largely affected by lattice distortions.

In high-entropy HE-LIC phases, the adjacent atoms at each lattice site often exhibit variations due to lattice distortion. Consequently, there are disparities in the configurations of neighboring atoms before and after ion migration, resulting in distinct local energies at each lattice site. When a Cl^- occupies a low-energy site, it becomes "trapped," reducing the likelihood of it leaving the site. Conversely, if the site is of high energy, there is a higher probability of the Cl^- returning to its original site, resulting in a failure of ion migration. Both scenarios contribute to a deceleration of the diffusion process. In contrast, traditional LIC typically exhibit consistent local atomic configurations before and after entering sites, thus avoiding pronounced hysteresis in diffusion. Visually, the variation of site energy makes the distribution of chlorine in HE-LIC become locally aggregated and disconnected compared to LIC (**Figure R21**). Additionally, high-entropy HE-LIC consist of multiple constituent elements, each with its own diffusion rate. The macroscopic diffusion behavior of these elements is the result of coordinated diffusion among the diverse elements. Consequently, the sluggish diffusion of elements in high-entropy HE-LIC significantly restricts the overall diffusion rate. The sluggish diffusion effect contributes to enhanced phase stability and oxidation stability in high-entropy HE-LIC. Commonly, high-entropy materials display enhanced hardness due to solid-solution strengthening, as well as good oxidation and corrosion resistance due to their sluggish diffusion, which limits penetration of oxygen and other species^{16,17}.

Figure R21. The probability density isosurface of chloride ions of the supercell based on BVEL: **a** HE-LIC, **b** LIC.

[Ref. R16] Sarkar A, *et al.* High-Entropy Oxides: Fundamental Aspects and Electrochemical Properties. *Adv. Mater.* **31**, e1806236 (2019).

[Ref. R17] Oses C, Toher C, Curtarolo S. High-entropy ceramics. *Nat. Rev. Mater.* **5**, 295-309 (2020).

4. While the HE-LIC is undoubtedly interesting, it remains challenging to identify a broader significance in designing advanced electrolyte compositions through this work. Further consideration for publication in a high-impact journal like Nature Communications should depend on addressing these comments.

Reply to the Reviewer: We thank the reviewer for giving us the opportunity to elucidate the significance of high-entropy halide solid electrolytes.

Our study represents a pioneering effort in exploring the intricate relationship between the alteration of configurational entropy and the consequential alterations in structure and electrochemical properties within halide solid electrolytes. Moreover, this study provides a comprehensive elucidation of the specific impact of local lattice distortion within high entropy structures through meticulous characterizations. As a result, a high-entropy halide solid electrolyte with exceptional ionic conductivity and remarkable electrochemical stability has been successfully developed. To the best of our knowledge, this is the first report of high-entropy materials for improving the electrochemical stability of halide SEs. This work holds immense significance for the progress of high-entropy materials in the field of ASSBs and expands our understanding of the fundamental principles governing solid electrolytes.

REVIEWERS' COMMENTS

Reviewer #1 (Remarks to the Author):

The overall requests raised by the reviewers have been very well addressed by the authors, and given by the work, this manuscript is suitable for publication in Nature Communications.

Reviewer #2 (Remarks to the Author):

The authors have incorporated all the required revisions and addressed ambiguities in the initial version of the manuscript. The manuscript can be accepted for publication in its current form.

Reviewer #3 (Remarks to the Author):

The authors have addressed the reviewer's concerns, improving the overall quality of the manuscript. The reviewer suggests including the XRD and electrochemical properties obtained from high-entropy solid electrolytes with cheap components to the revised Supporting Information. Additionally, the authors are needed to provide a comprehensive discussion on high-entropy solid electrolytes with cost-effective components in the main manuscript. Based on the improvements made, the reviewer recommends the publication of this manuscript in Nature Communications.

Itemized list of responses to the reviewers' report

(Black italics: Reviewers' report; Blue type: Reply to the Reviewer)

Reviewer #1: The overall requests raised by the reviewers have been very well addressed by the authors, and given by the work, this manuscript is suitable for publication in Nature Communications.

Reply to the Reviewer: We greatly thank the reviewer for taking time to review our manuscript. We appreciate your thorough review and constructive comments on our work.

Reviewer #2: The authors have incorporated all the required revisions and addressed ambiguities in the initial version of the manuscript. The manuscript can be accepted for publication in its current form.

Reply to the Reviewer: We are very grateful to the reviewer for the insightful remarks and suggestions in improving our work. We also appreciate your time and efforts in reviewing our work.

Reviewer #3: The authors have addressed the reviewer's concerns, improving the overall quality of the manuscript. The reviewer suggests including the XRD and electrochemical properties obtained from high-entropy solid electrolytes with cheap components to the revised Supporting Information. Additionally, the authors are needed to provide a comprehensive discussion on high-entropy solid electrolytes with cost-effective components in the main manuscript. Based on the improvements made, the reviewer recommends the publication of this manuscript in Nature Communications.

Reply to the Reviewer: We thank the reviewer for his/her overall summaries and comments of our work. We also appreciate your recommendation for the publication of our work.

Per the kind suggestions of the reviewer, we have included the XRD and electrochemical properties obtained from high-entropy solid electrolytes with cheap components as **Supplementary Figure 21, 22, 23**, as well as a discussion on high-entropy solid electrolytes with cost-effective elements in the manuscript.